# The Discovery of Potential Repellent Compounds for *Zeugodacus cucuribitae* (Coquillett) from Six Non-Favored Hosts

**DOI:** 10.3390/ijms26146556

**Published:** 2025-07-08

**Authors:** Yu Fu, Yupeng Chen, Yani Wang, Xinyi Fu, Shunda Jin, Chunyan Yi, Xue Bai, Youqing Lu, Wang Miao, Xingyu Geng, Xianli Lu, Rihui Yan, Zhongshi Zhou, Fengqin Cao

**Affiliations:** 1School of Tropical Agriculture and Forestry, Hainan University, Haikou 570100, China; 18189853537@163.com (Y.F.); ni0x0ni@163.com (Y.W.); fxy18289484592@163.com (X.F.); 17776994337@163.com (Y.L.); 18035319101@163.com (W.M.); 13693906876@163.com (X.G.); 18869855967@163.com (X.L.); ryan1@hainanu.edu.cn (R.Y.); 2Institute of Systems Medicine and Health Sciences, Hong Kong Baptist University, Hong Kong SAR 999077, China; chenyupeng@hkbu.edu.hk; 3School of Chinese Medicine, Hong Kong Baptist University, Hong Kong SAR 999077, China; 4School of Life Sciences, Sanya Institute of Henan University, Sanya 572025, China; shunda_jin@163.com; 5National Key Laboratory of Cotton Bio-Breeding and Integrated Utilization, Henan Joint International Laboratory for Crop Multi-Omics Research, School of Life Sciences, Henan University, Kaifeng 475000, China; 6Key Laboratory of Integrated Pest Management of Southwest Crops, Institute of Plant Protection, Sichuan Academy of Agricultural Sciences, Chengdu 610066, China; yichunyan2022@163.com; 7State Key Laboratory of Rice Biology and Breeding, Ministry of Agricultural and Rural Affairs Key Laboratory of Molecular Biology of Crop Pathogens and Insects, Key Laboratory of Biology of Crop Pathogens and Insects of Zhejiang Province, Institute of Insect Sciences, Zhejiang University, Hangzhou 310058, China; xuebai@zju.edu.cn; 8National Nanfan Research Institute, Chinese Academy of Agricultural Sciences, Sanya 572019, China

**Keywords:** *Zeugodacus cucuribitae* (Coquillett), olfactory genes, network pharmacology, molecular docking, insect repellent

## Abstract

*Zeugodacus cucuribitae* (Coquillett) (*Z. cucuribitae*) is a global extremely invasive quarantine pest which has a wide host range of fruits and vegetables. At present, there are a few control measures for *Z. cucuribitae*, and deltamethrin and avermectin are commonly used. Among the hosts of *Z. cucuribitae*, *Luffa acutangular*, *Luffa cylindrica*, *Sechium edule*, *Brassica oleracea* var. *botrytis*, *Musa nana*, and *Fragaria × ananassa* are non-favored hosts. However, it is still not clear why these hosts are non-favored and whether there are any repellent components of *Z. cucuribitae* in these hosts. In this study, the components of these six hosts were collected from the literature, and the genes of odor and chemical sensation were determined from the genome of *Z. cucuribitae*. After the potential relationships between these components and genes were determined by molecular docking methods, the KEGG and GO enrichment analysis of these genes was conducted, and a complex network of genes vs. components vs. Kegg pathway vs. GO terms was constructed and used to select the key components for experiments. The results show that oleanolic acid (1 mg/mL, 0.1 mg/mL, and 0.01 mg/mL), rotenone (1 mg/mL, 0.1 mg/mL, and 0.01 mg/mL), and beta-caryophyllene oxide (1 mg/mL, 0.1 mg/mL, and 0.01 mg/mL) had a significant repellent effect on *Z. cucuribitae*, and three components, rotenone (1 mg/mL and 0.1 mg/mL), echinocystic acid (1 mg/mL, 0.1 mg/mL, and 0.01 mg/mL), and beta-caryophyllene oxide (1 mg/mL, and 0.1 mg/mL) had significant stomach toxicity in *Z. cucuribitae*. Furthermore, a complex signaling pathway was built and used to predict the effect of these components on *Z. cucuribitae*. These components probably play roles in the neuroactive ligand–receptor interaction (ko04080) and calcium signaling (ko04020) pathways. This study provides a reference for the prevention and control of *Z. cucuribitae* and a scientific reference for the rapid screening and development of new pest control drugs.

## 1. Introduction

*Zeugodacus cucuribitae* (Coquillett) (*Z. cucuribitae*) is a destructive agricultural pest and is distributed across a range of climatic regions, such as Central and East Asia and Oceania [1]. Due to its vast adaptability, high reproduction potential, and invasion capacity, it has been the subject of a worldwide pest management program [2]. It has widely invaded many crops and brought about serious losses to agricultural production. *Z. cucuribitae* can damage a wide variety of fruits and vegetables, including *Luffa acutangular* (*L. acutangular*), *Luffa cylindrica* (*L. cylindrica*), and *Sechium edule* (*S. edule*) [3,4]. At present, chemical insecticides, such as cypermethrin, fenvalerate, and abamectin, were commonly used in the field to reduce the damage caused to fruits and vegetables by *Z. cucuribitae* [5]. However, conventional methods in controlling *Z. cucuribitae* have some limitations, such as environmental pollution and a small number of physical trappings [5,6,7,8]. Therefore, there is an urgent need to find some effective and environmental protective components to control *Z. cucuribitae*.

As reported, many plants have evolved various methods to resist insects, such as secondary metabolites. Attacks by *Bruchus pisorum* would induce *Pisum sativum* to produce an isoflavone compound called pisatin, which is an insect resistance substance produced by plants and enables resistance against the invasion of *B. pisorum* [9,10]. The feeding of *Spodoptera exigua* would induce the production of momilactone A and momilactone B in *Oryza sativa* L. and could achieve a defense effect against *Spodoptera exigua* [11]. At the same time, natural compounds extracted from plants have been reported to have good application prospects in pest control. For example, volatile substances and secondary metabolites in some plant extracts have significant repellent effects on *Bactrocera dorsalis* (Hendel) (*B. dorsalis*) [12,13]. These metabolites usually have high safety and environmental protection and have little impact on non-target organisms. Secondly, plant-derived compounds have been proven not to easily cause pests to develop drug resistance and have effectively been used for pest control for a long time [14,15].

Plants of the Cucurbitaceae family are the main hosts of *Z. cucuribitae*. However, *L. acutangular*, *L. cylindrica*, and *S. edule* belong to Cucurbitaceae and are the relatively non-favored hosts of *Z. cucuribitae*. Compared with other Cucurbitaceae plants, the visit rate and oviposition rate of these species by *Z. cucuribitae* are lower [16,17,18]. In addition, some non-Cucurbitaceae plants, such as *Brassica oleracea* var. *botrytis* (*B. oleracea*), *Musa nana (M. nana*), and *Fragaria × ananassa* (*F. ananassa*), are relatively less attractive to *Z. cucuribitae*. The relative host performance and fecundity of *Z. cucuribitae* on these species is significantly lower than that on other species [16,19]. However, there are still no reports about whether there are some compounds that have a repellent effect on *Z. cucuribitae*. The main purpose of this study was to discover the compounds of host plants with repellent effects and a low damage rate against *Z. cucuribitae*.

Insects’ perception of chemical compounds is a diverse and complex process. According to previous reports, insects have some compound receptor proteins, including an olfactory receptor (OR), odor binding protein and general odorant-binding proteins (OBPGOBP), gustatory receptor (GR), pheromone-binding protein (PBP), ionotropic receptor (IR), and sensory neuron membrane protein (SNMP) [20]. For example, OR, a complex family of membrane protein receptors, was responsible for olfactory perception and communication in most insects [21,22]. These proteins were widely used in the screening of compounds to control pests [23,24].

The combination of network pharmacology and molecular docking has been widely used in the identification of effective drugs in the medical field [25,26,27] and is gradually being applied in the study of pest control [28,29,30]. These integrated approaches could accelerate the development of pesticides, disease-resistant breeding, and the screening of natural products. For example, molecular docking has been used to screen the attractive compounds of *Z. cucuribitae* and to screen the insecticidal metabolic compounds of *Nephrolepis exaltata* extract against Culicidae [31].

In this study, the potential repellent compounds of *Z. cucuribitae* in six non-favored hosts were screened out mainly based on network pharmacology, molecular docking technology, and behavior determinations. By screening the genes of *Z. cucuribitae* genome, a number of proteins related to the perception of chemical compounds of *Z. cucuribitae* were identified. These genes were combined with the compounds contained in the metabolic compounds of the six hosts, molecular virtual screening was carried out to obtain potential repellent compounds, and a genes vs. compounds vs. Kegg pathways vs. GO terms (CPPG) network was constructed. The effective repellent compounds were finally screened out with the degree value in this network. This study proposes an effective method in calculating and screening the compounds that have potential effects on *Z. cucuribitae*. Furthermore, oleanolic acid, rotenone, and beta-caryophyllene oxide were proven to have significant repellent effects on *Z. cucuribitae* with two-way selection tests. Rotenone and beta-caryophyllene oxide were proven to have the significant stomach toxicity of *Z. cucuribitae* with stomach toxicity tests. These results provide a new basis for the development of a novel insecticide to control *Z. cucuribitae* and provide a scientific reference in the prevention and management of pests.

## 2. Results

### 2.1. Identification and Enrichment of Olfactory Sensory Genes in Z. cucuribitae

#### 2.1.1. Phylogenetic Analysis

The phylogenetic trees of *Z. cucurbitae*, including PBP, SNMP, OBP/GOBP, GR, OR, and IR genes, were constructed with R software (Version 4.3.1) after the identification of these genes from its genome *collected from the NCBI database*, *an* acknowledged database (Appendix A). The genes of *Z. cucurbitae* were clustered together with homologous genes from other Diptera insects, such as *B. dorsalis* and *Musca domestica* (*M. domestica*), demonstrating the accuracy of the odor-sensing genes of *Z. cucurbitae*. For example, in the phylogenetic analysis of IR genes (Figure 1), genes *BcucIR25a* and *BcucIR21a* from *Z. cucurbitae* and genes *BdorIR25a* and *BdorIR21a* from *B. dorsalis* formed a cluster with a short branching length, indicating high homology with the defined IR genes of *B. dorsalis*.

#### 2.1.2. KEGG and GO Enrichment

In the KEGG and GO enrichment analysis of the olfactory sensory genes of *Z. cucurbitae*, 16 pathways and 20 GO terms were identified (Figure 2A,B). The results show that the odor recognition gene *LOC105211704* was enriched in 16 Kegg pathways, including Systemic Lupus Erythematosus [Br: ko05322], Nicotine Addiction [Br: ko05033], Long-term Potentiation [Br: ko04720], Ion Channels [Br: ko04750], and Glutamatergic Synapse [Br: ko04724]. Previous studies have shown that Ion Channels [Br: ko04750] could affect the odor perception of the pea aphid *Acyrthosiphon pisum* to the host [32], and Glutamatergic Synapse [Br: ko04724] was related to neuronal development in *Drosophila melanogaster* and nerve sensation [33]. In addition, a total of 120 genes were enriched into 20 GO terms, including the sensory perception of chemical stimuli [GO:0007606] and the sensory perception of smell [GO:0007608]. According to studies, the sensory perception of smell [GO:0007608] has been shown to be closely related to OR and IR in *Locusta migratoria* [34].

#### 2.1.3. Metabolite Analysis of *L. acutangular*, *L. cylindrica*, *S. edule*, *B. oleracea*, *M. nana*, and *F. ananassa*

A total of 165 compounds of the six non-favored hosts, *L. acutangular*, *L. cylindrica*, *S. edule*, *B. oleracea*, *M. nana*, and *F. ananassa*, were collected from the literature. Among these 165 compounds, 55, 46, 10, 27, 57, and 24 compounds were from *L. acutangular*, *L. cylindrical*, *S. edule*, *B. oleracea*, *M. nana*, and *F. ananassa*, respectively (Table 1).

### 2.2. Molecular Docking

To identify whether the compounds of six non-favored hosts could effectively bind to the odorant proteins of *Z. cucurbitae*, molecular docking was performed on 166 compounds, and the proteins were translated by 196 olfactory sensory genes. A total of 239,475 relationships were obtained from the dockings.

The molecular docking information was evaluated with binding affinity, and the effective binding affinity ranged from −5.00 kcal/mol to −18.00 kcal/mol. The results show that A total of 160 compounds had affinity values equal to or lower than −5.00 kcal/mol, indicating that these compounds could successfully combine with the proteins of olfactory sensory genes of *Z. cucurbitae*. A total of 154 compounds had affinity values equal to or lower than −6.00 kcal/mol, and 40 compounds had affinity values equal to or lower than −10.00 kcal/mol. These 40 compounds had particularly strong binding capabilities and can be considered for further use.

Among the 196 genes encoding the proteins of olfactory sensory genes (Figure 3A), the average affinities of all genes were equal to or lower than −5.00 kcal/mol (Figure 3B), indicating that these compounds had strong affinity with the proteins of olfactory sensory genes of *Z. cucurbitae*.

Furthermore, according to the docking results, the affinity values of maslinic acid 3-O-β-D-glucoside, lucyoside H, and lucyoside G were the lowest, indicating that among all collected compounds, these three compounds had the strongest binding capabilities with the odor-sensing genes. Additionally, the genes *LOC105218953*, *LOC105209256*, and *LOC105215418* had the lowest affinity in all docking results of olfactory sensory genes, suggesting that these genes might be more sensitive to the compounds of the six hosts. *Z. cucurbitae* was likely to recognize harmful metabolites produced by the hosts through these genes and produce repellent or stomach toxicity effects.

The most effective combinations in each host are shown in Figure 3C–H. The maslinic acid 3-O-b-D-glucoside existing in *L. acutangular* could effectively bind with the gene *LOC105217972* with a binding energy of −9.28 kcal/mol (Figure 3C). The echinocystic acid existing in *L. cylindrica* could effectively bind with the gene *LOC105218953* with a binding energy of −11.71 kcal/mol (Figure 3D). The isorhoifolin existing in *S. edule* could effectively bind with the gene *LOC105215418* with a binding energy of −11.16 kcal/mol (Figure 3E). The kaempferol existing in *M. nana* could effectively bind with the gene *LOC105209256* with a binding energy of −11.80 kcal/mol (Figure 3F). The rutin existing in *B. oleracea* could effectively bind with the gene *LOC105209182* with a binding energy of −6.98 kcal/mol (Figure 3G). The damascenone existing in *F. ananassa* could effectively bind with the gene *LOC105217972* with a binding energy of −7.73 kcal/mol (Figure 3H).

According to reports, maslinic acid 3-O-b-D-glucoside was transformed by maslinic acid, which had a potential effect on pest control [71]. Echinocystic acid was isolated from *Eclipta prostrate* and played an important role in insect control by inhibiting the activation of NF-κB [72]. Isorhoifolin is a flavonoid, which played an important role in insect prevention and control by affecting the OR and GR of insects [73]. Rutin significantly reduced the feeding behavior and reproductive ability of *Bemisia tabaci* by regulating the defense mechanism of plants [74]. Rutin and quercetin could also significantly change the feeding behavior of aphids and reduce the damage on plants [75]. Damascenone, a pheromone, was released through modified cellulose nanocrystals to attract and control *Sogatella furcifera* [76].

### 2.3. Construction of CPPG Networks and Screening of Core Functional Compounds

The CPPG network was constructed based on the results of molecular docking and the KEGG and GO enrichment of olfactory sensory genes. The CPPG network contained 356 nodes, including 160 compounds, 196 genes, 16 pathways, and 20 GO terms (Figure 4). To more clearly illustrate the interactions between nodes, the top 30 compound nodes, top 30 gene nodes by degree, KEGG pathway nodes, and GO term nodes were selected to construct the CPPG network (Figure 4, Appendix A). Through an analysis of the CPPG network, the ranking of the 165 compounds from the host plants after docking with the odor recognition genes of *Z. cucurbitae* could be obtained. Futhermore, five compounds were randomly selected from the top 20 compounds which had the lowest degrees in the network and were used for further experiments of the two-way selections, and the gastric toxicity of oleanolic acid (C90), rotenone (C10), echinocystic acid (C91), diosmin (C72), and beta-caryophyllene oxide (C165) was finally determined. These compounds might play the most important role in *Z. cucurbitae*. Additionally, the nodes representing genes *LOC105218013*, *LOC105214272*, and *LOC105219487* had the highest degree, indicating the greatest interaction with each compound.

### 2.4. Behavior Determination of Z. cucuribitae

#### 2.4.1. Results of Two-Way Selection Experiments

The results of the two-way experiments show that oleanolic acid, rotenone, and beta-caryophyllene oxide had significant repellent effects on *Z. cucurbitae* at concentrations of 0.01 mg/mL, 0.1 mg/mL, and 1 mg/mL (Figure 5A,C,D). However, different concentrations of echinocystic acid and diosmin had no significant effect (Figure 5B,F). The repellent rate of 0.1 mg/mL rotenone could reach 86.67%, representing the most remarkable repellent effect on *Z. cucurbitae*.

#### 2.4.2. Results of Gastric Toxicity Experiments

The results of the gastric toxicity experiments show that the mortality of *Z. cucurbitae* in the beta-caryophyllene oxide treatment reached 63.33% and 53.33% at concentrations of 1 mg/mL and 0.1 mg/mL, respectively (Figure 5F), which were significantly higher than the control (21.67%). The mortality rate of *Z. cucurbitae* in the rotenone treatment reached 63.33% and 50.00% at concentrations of 1 mg/mL and 0.1 mg/mL, respectively (Figure 5H), indicating a significant toxic effect on *Z. cucurbitae*. Echinocystic acid had significant insecticidal activity at concentrations of 1 mg/mL, 0.1 mg/mL, and 0.01 mg/mL, with mortality rates of 75.33%, 55.33%, and 55.00%, respectively (Figure 5J). Among these, the highest mortality was observed at a concentration of 1 mg/mL, and echinocystic acid also had a significant repellent effect at all concentrations. However, different concentrations of diosmin did not show significant effects in the experiment (Figure 5G,I).

These compounds also had certain control effects on other insects. According to reports, beta-caryophyllene oxide isolated from plants had obvious repellent activity against *Tribolium castaneum* and *sitophilus granarius* L. [77,78] and exhibited considerable fumigation toxicity and repellent characteristics toward *Callosobruchus chinensis* [79]. In addition, oleanolic acid had significant larvicidal activity against *Aedes aegypti* L. [80] and the significant antifeedant activity against *spodoptera litura* F. [81]. Rotenone is a broad-spectrum insecticidal active ingredient, which has a cytotoxic effect on SL-1 cells and significantly inhibits the larval growth of *Spodoptera litura* [82,83]. Furthermore, rotenone could disrupt the energy metabolism and mitochondrial dysfunction of *Bombus terrestis* glial cells and damage intestinal peristalsis [84].

### 2.5. Predicted Results of Molecular Mechanisms

The results of the two-way selection experiments and gastric toxicity experiments verified the effective effects of oleanolic acid, rotenone, and beta-caryophyllene oxide on *Z. cucurbitae*. The control effects of these compounds on *Z. cucurbitae* were achieved through the joint action of genes and pathways. According to the complex network, the targets of the successful actions of these compounds could be identified. Through the constructed CGGP network, we identified the gene *LOC105217288*. After the KEGG enrichment of *LOC105217288*, it was found that these compounds were mainly enriched in two pathways, the neuroactive ligand–receptor interaction (ko04080) and calcium signaling (ko04020) pathways. By marking the activated genes in these two pathways, we determined a molecular mechanism of the influence of the compounds apigenin, testosterone propionate, and biochanin A on the behavior of *Z. cucurbitae* (Figure 6). These compounds further activate proteins located on the cell membrane by activating the gene *LOC105217288*, including the glutamate receptor ionotropic NMDA1 (GRIN), glutamate receptor 1 (GRI), and nicotinic acetylcholine receptor alpha-7 (nAChRα7). These targets further affect downstream pathways by activating calcium ions and calmodulin (CALM), such as the MAPK signaling pathway (ko04010), apoptosis (ko04210), and long-term depression (ko04730). Eventually, *Z. cucurbitae* exhibits avoidance behavior and even death. Reports have shown that neonicotinoids, such as imidacloprid, could activate neuroactive ligand–receptor interaction (ko04080), leading to the dysfunction of insect nervous system and subsequently causing oxidative stress, mitochondrial dysfunction, inflammatory reaction, and apoptosis [85]. By combining RNA-Seq with isobaric ectopic tags for relative and absolute quantitative (iTRAQ) analysis, it was found that the calcium signaling pathway (ko04020) plays an important role in the adaptation and toxic reaction of *Bursaphelenchus mucronatus* to the host [86].

## 3. Discussions

*Z. cucurbitae* is a destructive pest which can damage a wide variety of fruits and vegetables, including *L. acutangular*, *L. cylindrica*, and *S. edule*. Currently, chemical pesticides are the main method in controlling *Z. cucurbitae*. However, this method has obvious limitations, such as the limited effectiveness of deltamethrin and abamectin [3,4,7]. Previous studies have shown that plant metabolites had good control effects and economic value in pest control. For instance, azadirachtin isolated from *Azadirachta indica* was proven to effectively control pests [87,88]. Host plants also contain insect-resistant metabolites to defend against insect damage [89,90,91]. Long-term observations indicate that although *Z. cucurbitae* can harm *L. acutangular*, *L. cylindrica*, *S. edule*, *B. oleracea*, *M. nana*, and *F. ananassa*, the damage is not severe [16,92]. It has been speculated that both cucurbitaceous hosts, such as *L. acutangular*, *L. cylindrica*, and *S. edule*, and non-cucurbitaceous hosts, such as *B. oleracea*, *M. nana*, and *F. ananassa*, contain metabolites that adversely affect Z. cucurbitae, thereby reducing the damage caused by this pest. These potential metabolites are worthy of further exploration and study for the prevention and control of *Z. cucurbitae*.

The compounds of *L. acutangular*, *L. cylindrica*, *S. edule*, *B. oleracea*, *M. nana*, and *F. ananassa* were collected from the literature. Meanwhile, the olfactory sensory genes of *Z. cucurbitae* were gathered based on its genome. Molecular docking was then conducted between these compounds and the olfactory sensory genes. Moreover, the genes were enriched via the KEGG pathway and GO term analyses, and a CPPG network was constructed. Through these steps, important compounds that may have potential effects on *Z. cucurbitae* were identified.

In the two-way selection experiment, oleanolic acid (0.01 mg/mL, 0.1 mg/mL, and 1 mg/mL), rotenone (0.01 mg/mL, 0.1 mg/mL, and 1 mg/mL), and beta-caryophyllene oxide (1 mg/mL and 0.1 mg/mL) were successfully screened. In the two-way selection experiment, echinocystic acid (0.01 mg/mL, 0.1 mg/mL, and 1 mg/mL), rotenone (0.1 mg/mL and 1 mg/mL), and beta-caryophyllene oxide (1 mg/mL and 0.1 mg/mL) have significant repellent effects against *Z. cucurbitae*. In the gastric toxicity experiment, echinocystic acid (0.01 mg/mL, 0.1 mg/mL, and 1 mg/mL), rotenone (0.1 mg/mL and 1 mg/mL), and beta-caryophyllene oxide (0.1 mg/mL and 1 mg/mL) had significant gastric toxicity against *Z. cucurbitae*. Among these compounds, rotenone has been reported as a widely used botanical insecticide, which has various insecticidal activities, including neurotoxicity against *Spodoptera litura* and *Bombus terrestis* [82,83]. Beta-caryophyllene oxide has been widely reported to have significant toxicity against Sitophilus *Granarius* L. and *Callosobruchus chinensis* [77,78,79]. Oleanolic acid has been found to have had significant antifeedant activity against *Aedes aegypti* L. and *spodoptera litura* F. [80,81]. However, there is still a lack of studies of echinocystic acid in controlling pests, indicating that this compound is worth being further studied.

After verifying the effects of oleanolic acid, rotenone, and beta-caryophyllene oxide on *Z. cucurbitae*, the gene *LOC105217288* was identified through the KEGG enrichment of the gene set successfully docked with these compounds. Mechanism prediction was carried out after constructing a composite pathway, and it was found that *LOC105217288* is mainly involved in the neuroactive ligand–receptor interaction (ko04080) and calcium signaling (ko04020) pathways. In these pathways, the proteins glutamate receptor ionotropic NMDA1 (GRIN), glutamate receptor 1 (GRI), and nicotinic acetylcholine receptor alpha-7 (nAChRα7) were activated, affecting downstream proteins and pathways. In the composite pathway, calmodulin (CALM) was activated, which influenced pathways such as the MAPK signaling pathway (ko04010), apoptosis (ko04210), and long-term depression (ko04730). Neuroactive ligand–receptor interaction (ko04080) and calcium signaling (ko04020) pathways have been widely reported in insect research and are closely related to insect avoidance and death [85,86]. Additionally, the activation of these pathways could indirectly affect downstream pathways, such as the MAPK signaling pathway (ko04010), apoptosis (ko04210), and long-term depression (ko04730). Transcriptome analysis showed that the expression of the MAPK signaling pathway (ko04010) changed after *Bombyx mori* was parasitized by *Exorista japonica* and played an important role in the response of insects to parasitic stress [93]. Transcriptome analysis and PCR-RFLP analysis showed that the differential genes of *Aphis gossypii* and *Aedes aegypti*, after being treated with pyrethroid insecticides, were closely related to the genes in the apoptosis pathway (ko04210) [94,95].

## 4. Materials and Methods

### 4.1. Insect Rearing

*Z. cucuribitae* was reared in the Invasive Pest Laboratory, Hainan University (Haikou, China). The larvae of the *Z. cucuribitae* colony were provided with an artificial larval diet mixture of 50 g of yeast extract, 250 g of wheat bran powder, 50 g of sugar, 1 g of sodium benzoate, 50 g of paper, and 400 mL of water. Adults were fed artificial diets of a 3:1 ratio of sucrose/yeast extract. All experimental adults were maintained in cages (60 × 60 × 60 cm) at 27 ± 1 °C under a 16 h/8 h light/dark cycle at a relative humidity of 70 ± 5% [96].

### 4.2. Identification and Functional Enrichment of Odor-Sensing Genes from Z. cucuribitae

Odor-sensing genes sets, including the OBP, PBP, OR, IR, GR, and SNMP genes of *Z. cucuribitae*, were collected based on the genome data (ID: GCF028554725.1) which was obtained from the NCBI database (https://www.ncbi.nlm.nih.gov/, accessed on 16 January 2025). In this process, phylogenetic analysis of the odor-sensing genes of *Z. cucurbitae* was conducted to verify the reliability of the gene function [20]. A phylogenetic tree was constructed and visually displayed using the APE package and the ggTree package of R software (Version 4.3.1) (R Foundation for Statistical Computing, Vienna, Austria) [97].

Among the genetic data, the GR data set contains 29 sequences from several species, such as *Z. cucuribitae*, *Drosophila melanogaster* (*D. melanogaster*), and *B. dorsalis*. The IR data set contains 8 sequences from several species, such as *Z. cucuribitae*, *D. melanogaster*, and *M. domestica*. The OBPGOBP data set contains 49 sequences from several species, such as *Z. cucuribitae*, *Apis Cerana* (*A. Cerana*), and *Drosophila Simulans*. The OR data set contains 95 sequences from several species, such as *Z. cucuribitae*, *D. melanogaster*, and *A. cerana*. The PBP data set contains 7 sequences from several species, such as *Z. cucuribitae*, *B. dorsalis*, and *M. domestica*. The SNMP data set contains 8 sequences from several species, such as *Z. cucuribitae*, *B. dorsalis*, and *M. domestica*. KEGG enrichment and GO enrichment were performed on odor-related protein targets (*p* < 0.05). The Cluster Profiler package of R software (Version 4.3.1) (R Foundation for Statistical Computing, Vienna, Austria) was used for enrichment analysis, and the ggplot2 package was used for visualization [98].

### 4.3. Acquisition of Compound Models of L. acutangular, L. cylindrica, S. edule, B. oleracea, M. nana, and F. ananassa and Protein Models of Z. cucuribitae

The compounds from six host plants were collected from published studies and integrated into the public database PubChem (https://pubchem.ncbi.nlm.nih.gov, accessed on 20 February 2025). Three-dimensional structures of compounds were collected from the PubChem database in the sdf format and converted into the mol2 format using OpenBABEL software (Version 3.1.1) (Open Babel development team, Cambridge, MA, USA) for molecular docking.

Three-dimensional protein structural models corresponding to all odor-sensing genes were predicted based on the d SWISS-MODEL (https://swissmodel.expasy.org/interactive, accessed on 5 March 2025) and saved in pdb format. Kegg pathways and GO terms related to odor-sensing genes were collected using KEGG and GO enrichment. The Cluster Profiler package of R software (Version 4.3.1) (R Foundation for Statistical Computing, Vienna, Austria) was used for enrichment analysis, and the ggplot2 package was used for visualization.

### 4.4. CPPG Networks and Screening of Core Functional Compounds

#### 4.4.1. Molecular Docking of Olfactory Sensory Genes and Metabolic Compounds

All docking relationships between compounds of hosts and odor-sensing genes of *Z. cucuribitae* were obtained by molecular docking with AutoDock-VINA (Version 1.1.2) (The Scripps Research Institute, La Jolla, CA, USA). In order to evaluate the results of molecular docking, the affinity value from molecular docking was used to evaluate the effectiveness of protein binding with chemical compounds. A lower affinity value represents better binding energy. When the affinity value was equal to or less than −5 kcal/mol and greater than or equal to −18 kcal/mol, it was considered that the protein can effectively bind with chemical compounds [99,100]. Finally, PyMOL (Version 2.5.4) (Schrödinger, Inc., New York, NY, USA) software was used for visualization.

Data analysis was conducted using Poisson distribution for a non-normal distribution in a generalized linear model. Then, analysis of variance (ANOVA) and multiple comparisons were used for significance analysis. Finally, a histogram was created using the ggplot2 package. The data were analyzed using the R software (version 4.3.1) (R Foundation for Statistical Computing, Vienna, Austria) and multcomp, glm, emmeans, lsmeans, ggsignif, and ggplot2 packages.

#### 4.4.2. Construction of CPPG Networks and Screening of Core Functional Compounds

Network visualization was carried out using Cytoscape (Version 3.9.1) (Cytoscape Consortium, Seattle, WA, USA). Based on the results of molecular docking, a CPPG network was constructed with metabolites from *L. acutangular*, *L. cylindrica*, *S. edule*, *B. oleracea*, *M. nana*, and *F. ananassa*, olfactory sensory genes of *Z. cucurbitae*, KEGG pathways, and GO terms as nodes. To clarify the relationships between different nodes, the degree of each node was used as an index to judge the importance of nodes. The degree was calculated as the product of the number of successful docking results and the average binding energy for that node. The higher the degree was, the more important the node was in the network, and the node played a more significant role in the comprehensive effect relationships among genes, compounds, KEGG pathways, and GO terms [29,30,101].

### 4.5. Effect of Core Compounds on Behavior of Z. cucuribitae

#### 4.5.1. Two-Way Selection Experiment

A two-way selection experiment was conducted in an independent and ventilated laboratory. The behavior measurement laboratory was equipped with an exhaust fan for ventilation, maintained in a dark environment with a temperature of 25 ± 2 °C, a relative humidity of 70 ± 5%, and a photoperiod of 16 h:8 h. Each compound to be verified was dissolved in 5 µL of dimethyl sulfoxide (DMSO, RT, 99%), and distilled water was added. The concentration of DMSO used in the cell experiments needed to be less than 0.1% [29]. All *Z. cucurbitae* were starved for 24 h. Each compound was prepared into solutions of 0.01 mg/mL, 0.1 mg/mL, and 1 mg/mL, respectively. Screen cages (60 cm × 60 cm × 60 cm) were placed in the behavior measurement laboratory, and two thick slices of *Cucurbita pepo* (*C. pepo*) (radius ≈ 5 cm, thickness ≈ 2 cm) were placed in each cage. One slice was evenly coated with the treatment group liquid on the surface, and the other was coated with a DMSO solution of the same concentration as the control. Fifty adult *Z. cucurbitae* (male/female = 1:1) were placed in each cage, and the numbers on the *C. pepo* slices regarding the treatment and control were determined after 10 min. To reduce the possibility of interference between treatments as much as possible, the avoidance behavior test was carried out in an independent room, and each treatment was repeated 3 times.$$repellent\;rate=\frac{Number\;of\;Control-Number\;of\;Treatment}{Number\;of\;Control+Number\;of\;Treatment}\times 100\%$$

#### 4.5.2. Gastric Toxicity Experiment

Based on the methods of the two-way selection experiment described above, the compounds to be verified were prepared into solutions of 0.01 mg/mL, 0.1 mg/mL, and 1 mg/mL. The experiment was also conducted in an independent laboratory. An amount of 10 mL of each treatment group liquid of the solution was poured into a 250 mL conical flask (DaLong Experimental Instrument, Beijing, China), shaken well, and then slowly rotated for 1 min to form a uniform film on the flask wall. When the flask was dry, 30 adult *Z. cucurbitae* (male/female = 1:1) were placed in a conical flask coated with the drug film for 24 h and then transferred to a clean 250 mL conical flask (DaLong Experimental Instrument, Beijing, China) for observation. Each treatment was repeated three times.$$mortality\;rate=\frac{Number\;of\;dead\;insects}{Number\;of\;test\;insects}\times 100\%$$

### 4.6. Predicted Mechanism of Core Compounds

After verification using the two-way selection experiment and gastric toxicity experiment, genes of verified core components were collected in CPPG network. Furthermore, the KEGG enrichment results linked to genes were also found and collected. The maps of these pathways were obtained from the KEGG database, and the positions of these genes in the pathways were accurately marked. Combined with the genes and products of the upstream and downstream pathways of these genes, the action mechanism of composite pathways was predicted. Furthermore, in combination with the positions of marked genes, the influence of metabolic reactions or signal transduction events on the operation of the entire pathway was discussed.

## 5. Conclusions

In this study, the combination of network pharmacology and molecular docking technology was used to screen the components that have a repellent effect on *Z. cucurbitae*. This method comprehensively analyzes the relationship compounds of six non-favored hosts (*L. acutangular*, *L. cylindrica*, *S. edule*, *B. oleracea*, *M. nana*, and *F. ananassa*) and odor-sensing genes of *Z. cucurbitae* by constructing a CPPG network. Through this method, the compounds rotenone, beta-caryophyllene oxide, and echinocystic acid were successfully identified to exhibit significant repellent effects. These compounds were first found to have significant insecticidal activities against *Z. cucurbitae*. This study lays a solid foundation for further studies on the repellent mechanisms of *Z. cucurbitae* and the development of new and efficient control strategies. Future research will further explore the mechanisms, optimization, and applications of these metabolites to provide a scientific reference for the effective prevention and control of *Z. cucurbitae*.

## Figures and Tables

**Figure 1 ijms-26-06556-f001:**
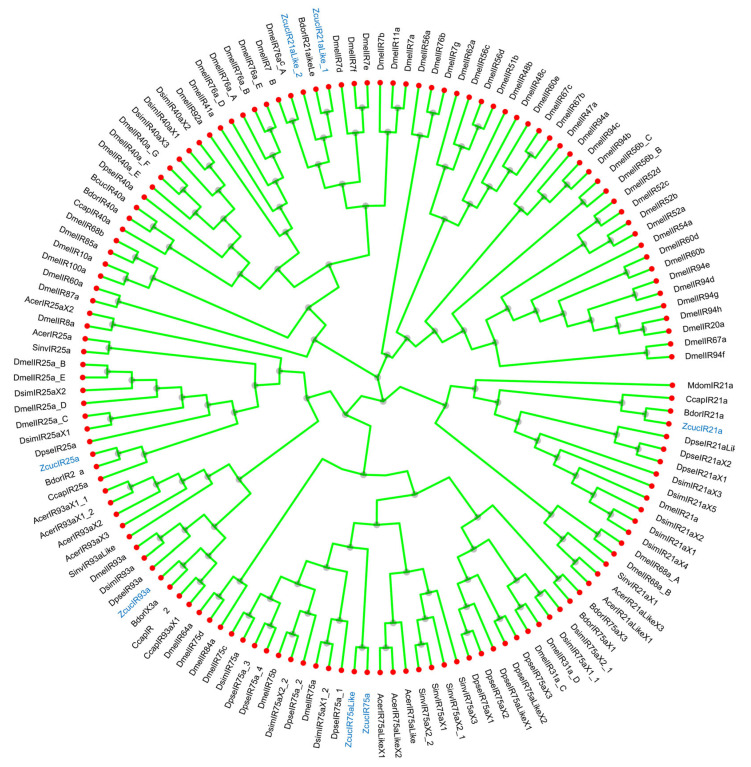
Construction of phylogenetic analysis of IR genes based on gene sequences of *Z. cucuribitae* and several Diptera insects. Genes from *Z. cucuribitae* are marked in blue. Notes: Acer: *Apis Cerana*, Zcuc: *Zeugodacus Cucurbitae*, Bdor: *Bactrocera Dorsalis*, Bole: *Bactrocera Oleae*, Dmel: *Drosophila Melanogaster*, Mdom: *Musca Domestica*, Dana: *Drosophila Ananassae*, Dsim: *Drosophila Simulans*.

**Figure 2 ijms-26-06556-f002:**
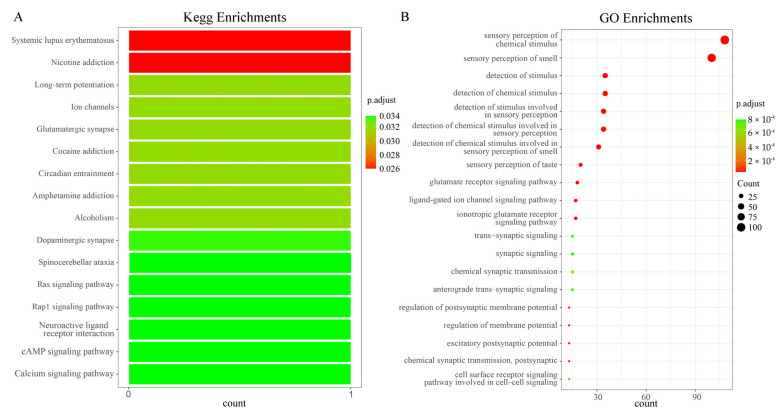
KEGG and GO enrichment. (**A**) KEGG enrichment; (**B**) GO enrichment.

**Figure 3 ijms-26-06556-f003:**
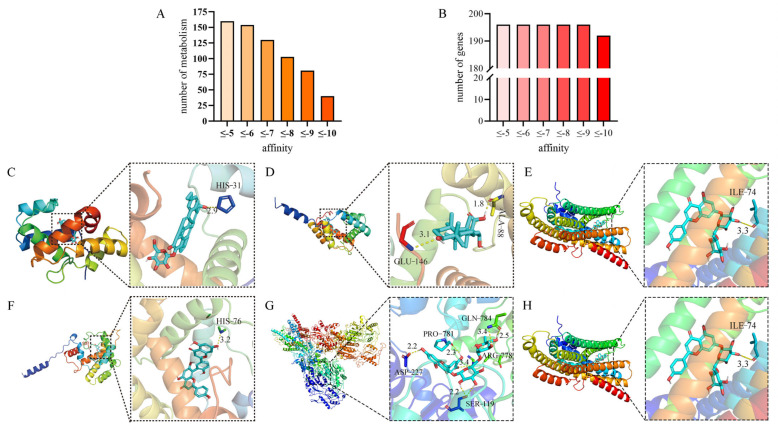
A visualization of the docking of olfactory sensory genes with the compounds of hosts. (**A**) The numbers of compounds with different affinities between −5 kcal/mol and 18 kcal/mol were counted. (**B**) The numbers of genes with different affinities between −5 kcal/mol and −18 kcal/mol were counted. (**C**) The prediction results of molecular docking between maslinic acid 3-O-b-D-glucoside and the *LOC105217972* gene of Luffa cylindrica. (**D**) The prediction of the combination of the metabolic compound echinocystic acid and the gene *LOC105218953* of Luffa cylindrica. (**E**) The prediction of the combination of isorhoifolin and the gene *LOC105215418*. (**F**) The prediction of the combination of kaempferol, a metabolic compound of banana, and the gene *LOC105209256*. (**G**) The prediction of the combination of rutin, a metabolic compound of cauliflower, and the gene *LOC105209182*. (**H**) The prediction of the combination of damascenone, a metabolic compound of strawberry, and the gene *LOC105217972*. Notes: The molecular docking of Ligand X with Protein Receptor Y is shown. The figure illustrates the binding mode of Ligand X (shown in stick representation, colored by atom type, with carbon shown in blue and oxygen shown in red) within the active site of Protein Receptor Y (shown in cartoon representation, colored by secondary structure). Ligand X (stick representation) is shown interacting with key residues (labeled) of the receptor (cartoon representation). Hydrogen bonds are depicted as dashed lines.

**Figure 4 ijms-26-06556-f004:**
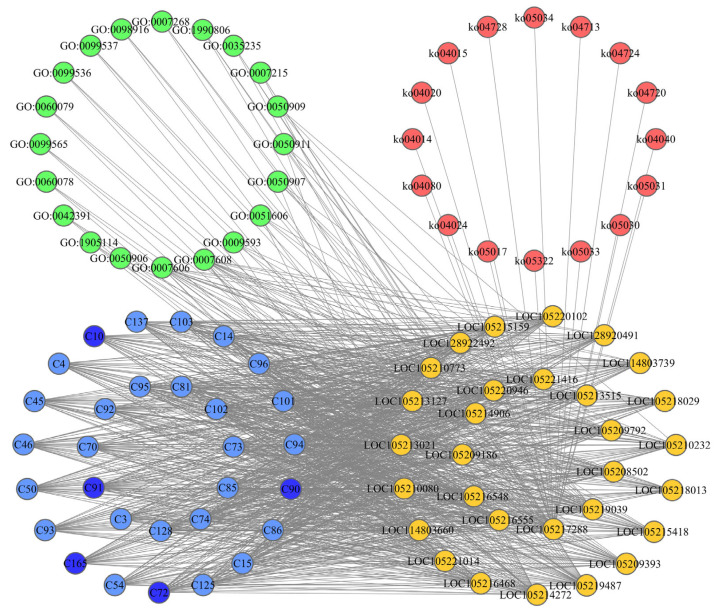
A complex network node diagram of the compounds of *L. acutangular*, *L. cylindrica*, *S. edule*, *B. oleracea*, *M. nana*, and *F. ananassa* vs. Z. *cucuribitae* olfactory sensory genes vs. Kegg pathway vs. GO term. The core compounds are presented in blue, the *Z. cucuribitae* olfactory sensory genes are presented in yellow, the kegg pathway is presented in green, and the GO term is presented in red. The metabolites selected for the two-way selection test and gastric toxicity test are shown in dark blue.

**Figure 5 ijms-26-06556-f005:**
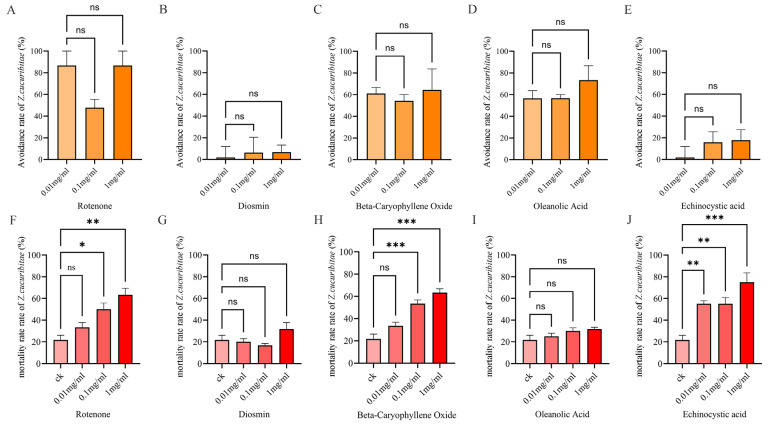
Avoidance rate and mortality rate of *Z. cucuribitae* under different concentrations of five metabolic compounds. (**A**) Avoidance rate of rotenone. (**B**) Avoidance rate of diosmin. (**C**) Avoidance rate of beta-caryophyllene oxide. (**D**) Avoidance rate of oleanolic acid. (**E**) Avoidance rate of echinocystic acid. (**F**) Mortality rate of rotenone. (**G**) Mortality rate of diosmin. (**H**) Mortality rate of beta-caryophyllene oxide. (**I**) Mortality rate of oleanolic acid. (**J**) Mortality rate of echinocystic acid. Note: ck means control check; ns means no significance between two compared groups. * represents significant difference at *p* < 0.05; ** represents significant difference at *p* < 0.01; *** represents significant difference at *p* < 0.001 compared to model group.

**Figure 6 ijms-26-06556-f006:**
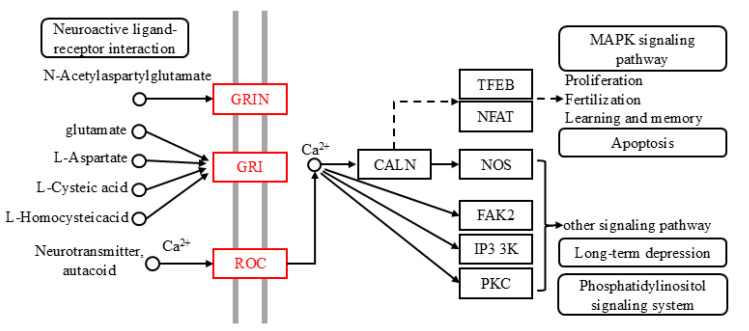
Mechanism deduction of effect of effective compounds on odor recognition of *Z. cucurbitae*. Red nodes represent activated genes. Notes: GRIN: glutamate receptor ionotropic; GRI: glutamate receptor 1; ROC: nicotinic acetylcholine receptor alpha-7; CALN: calmodulin; TFEB: transcription factor EB; NFAT: nuclear factor of activated T-cells, cytoplasmic 1; NOS: nitric-oxide synthase, brain; FAK2: focal adhesion kinase 2; IP3 3K: 1D-myo-inositol-triphosphate 3-kinase; PKC: classical protein kinase C alpha type.

**Table 1 ijms-26-06556-t001:** Analysis of *L. acutangular*, *L. cylindrica*, *S. edule*, *B. oleracea*, *M. nana*, and *F. ananassa*.

Name	ID	CAS	Host	References
cinncassiol E	C1		*L. acutangular*	[35]
hymenoxon	C2	57377-32-9	*L. acutangular*	[35]
1-(alpha-methyl-4-(2-methylpropyl)benzeneacetate)-beta-D-Glucopyranuronic acid	C3	115075-59-7	*L. acutangular*	[35]
cinncassiol C3	C4		*L. acutangular*	[35]
clerodin	C5	464-71-1	*L. acutangular*	[35]
nigakilactone E	C6	28360-79-4	*L. acutangular*	[35]
shogaol	C7	555-66-8	*L. acutangular*	[35]
maritimetin	C8	576-02-3	*L. acutangular*	[35]
gartanin	C9	33390-42-0	*L. acutangular*	[35]
rotenone	C10	83-79-4	*L. acutangular*	[35]
apiin	C11	26544-34-3	*L. acutangular*	[35]
galangin	C12	548-83-4	*L. acutangular*	[35]
hispidulin	C13	1447-88-7	*L. acutangular*	[35]
berbamunine	C14	485-18-7	*L. acutangular*	[35]
daphnoline	C15	479-36-7	*L. acutangular*	[35]
ellagic acid	C16	476-66-4	*L. acutangular*, *L. cylindrica*	[36,37]
chlorogenic acid	C17	202650-88-2	*L. acutangular*, *L. cylindrica*, *M. nana*, *B. oleracea*	[36,38,39,40]
sinapaldehyde	C18	4206-58-0	*L. acutangular*	[36]
mandelic acid	C19	90-64-2	*L. acutangular*	[36]
scopoletin	C20	92-61-5	*L. acutangular*	[36]
salicylic acid	C21	69-72-7	*L. acutangular*, *L. cylindrica*, *M. nana*, *B. oleracea*	[36,38,39,41]
gallic acid	C22	149-91-7	*L. acutangular*, *L. cylindrica*, *M. nana*, *B. oleracea*	[36,38,39,40]
4-hydroxycoumarin	C23	1076-38-6	*L. acutangular*	[36]
phenylacetic acid	C24	103-82-2	*L. acutangular*	[36]
guaiacol	C25	32994	*L. acutangular*	[36]
3,4-dihydroxyphenylacetic acid	C26	102-32-9	*L. acutangular*	[36]
hydroxycaffeic acid	C27		*L. acutangular*, *L. cylindrica*, *B. oleracea*, *F. ananassa*	[36,40,42,43]
2,5-dihydroxybenzoic acid	C28		*L. acutangular*, *L. cylindrica*, *M. nana*	[36,38,39]
4-ethylguaiacol-d5	C29		*L. acutangular*	[36]
3-(4-hydroxy-3-methoxyphenyl)propionic acid	C30	1135-23-5	*L. acutangular*	[36]
caffeic acid	C31	501-16-6	*L. acutangular*, *M. nana*	[36,39]
pinocembrin	C32	480-39-7	*L. acutangular*	[36]
benzoic acid	C33	65-85-0	*L. acutangular*	[36]
cirsimaritin	C34	6601-62-3	*L. acutangular*	[36]
eugenol	C35	97-53-0	*L. acutangular*	[36]
p-coumaric acid	C36	501-98-4	*L. acutangular*, *L. cylindrica*, *M. nana*, *B. oleracea*, *F. ananassa*	[36,38,39,40,43]
sakuranetin	C37	2957-21-3	*L. acutangular*	[36]
gardenin B	C38	2798-20-1	*L. acutangular*	[36]
chrysin	C39	480-40-0	*L. acutangular*, *M. nana*	[36,39]
scutellarein	C40	529-53-3	*L. acutangular*	[36]
tetramethylscutellarein	C41	1168-42-9	*L. acutangular*	[36]
mellein	C42	1200-93-7	*L. acutangular*	[36]
geraldone	C43	21583-32-4	*L. acutangular*	[36]
umbelliferone	C44	202-240-3	*L. acutangular*	[44]
lucyoside A	C45		*L. acutangular*	[44]
lucyoside J	C46	100156-31-8	*L. acutangular*	[44]
luteolin	C47	491-70-3	*L. acutangular*, *L. cylindrica*, *S. edule*, *M. nana*	[39,44,45,46]
apigenin	C48	520-36-5	*L. acutangular*, *M. nana*	[39,44]
diosmetin	C49	520-34-3	*L. acutangular*, *S. edule*	[44]
lucyoside G	C50		*L. acutangular*	[44]
13-trihydroxy-octadecenoic acid	C51		*L. acutangular*	[44]
lucyoside H	C52		*L. acutangular*	[44]
lucyoside I	C53	99543-11-0	*L. acutangular*	[44]
Maslinic acid 3-O-b-D-glucoside	C54	163634-06-8	*L. acutangular*	[44]
lucyin A	C55	152845-76-6	*L. acutangular*	[44]
4-hydroxybenzoic acid	C56	99-96-7	*L. cylindrica*	[38]
catechol	C57	120-80-9	*L. cylindrica*, *B. oleracea*	[38,40]
vanillic acid	C58	121-34-6	*L. cylindrica*, *M. nana*, *B. oleracea*	[38,39,40]
rutin	C59	153-18-4	*L. cylindrica*, *M. nana*, *B. oleracea*	[38,39,40]
ferulic acid	C60	537-98-4	*L. cylindrica*, *M. nana*, *B. oleracea*, *F. ananassa*	[38,39,40,43]
naringenin	C61	480-41-1	*L. cylindrica*, *M. nana*, *B. oleracea*	[38,39,40]
2-hydroxy-4-methylbenzaldehyde	C62	698-27-1	*L. cylindrica*	[47]
4-acetoxy-2-azetidinone	C63	28562-53-0	*L. cylindrica*	[47]
mahaleboside	C64		*L. cylindrica*	[47]
crotanecine	C65	5096-50-4	*L. cylindrica*	[47]
perlolyrine	C66	29700-20-7	*L. cylindrica*	[47]
dihydrocapsaicin	C67	19408-84-5	*L. cylindrica*	[47]
morindone	C68	478-29-5	*L. cylindrica*	[47]
4-aminosalicylic acid	C69	65-49-6	*L. cylindrica*	[38]
apigenin 7-glucuronide	C70	29741-09-1	*L. cylindrica*	[48]
kaempferide	C71	491-54-3	*L. cylindrica*,	[49]
diosmin	C72	520-27-4	*L. cylindrica* *S. edule*	[50,51]
5-hydroxy-2-(3-hydroxy-4-methoxyphenyl)-4-oxo-4H-1-benzopyran-7-yl 2-O-(6-deoxy-alpha-L-mannopyranosyl)-beta-D-glucopyranoside	C73	38665-01-9	*L. cylindrica*	[52]
eriodictyol-7-O-glucoside	C74	38965-51-4	*L. cylindrica*	[52]
quercetin	C75	117-39-5	*L. cylindrica*, *M. nana*, *B. oleracea*	[39,40,53]
myricetin	C76	529-44-2	*L. cylindrica*	[54]
cianidanol	C77	7295-85-4	*L. cylindrica*, *M. nana*, *B. oleracea*	[39,40,55]
hyperoside	C78	482-36-0	*L. cylindrica*	[56]
lespedin	C79	482-38-2	*L. cylindrica*	[57]
quercitrin	C80	522-12-3	*L. cylindrica*, *M. nana*	[39,58]
tiliroside	C81	20316-62-5	*L. cylindrica*	[59]
acacetin	C82	480-44-4	*L. cylindrica*, *M. nana*	[39,60]
saponarin	C83	20310-89-8	*L. cylindrica*	[60]
datiscin	C84		*L. cylindrica*	[61]
fortunellin	C85	20633-93-6	*L. cylindrica*	[61]
linarin	C86	480-36-4	*L. cylindrica*	[62]
vitexin	C87	3681-93-4	*L. cylindrica*, *S. edule*	[51,63]
vitexin 2″-O-rhamnoside	C88	64820-99-1	*L. cylindrica*	[64]
2-hydroxycinnamic acid,(2E)-	C89	614-60-8	*L. cylindrica*, *M. nana*	[39,64]
oleanolic acid	C90	508-02-1	*L. cylindrica*	[65]
echinocystic acid	C91	510-30-5	*L. cylindrica*	[66]
gypsogenin	C92	639-14-5	*L. cylindrica*	[67]
3-O-[beta-D-glucopyranosyl]-28-O-[alpha-L-rhamnopyranosyl-(1->2)-beta-D-glucopyranosyl]maslinic acid	C93	1268696-94-1	*L. cylindrica*	[68]
luteolin 7-rutinoside	C94	20633-84-5	*S. edule*	[51]
luteolin 7-O-glucoside	C95	1268798	*S. edule*, *M. nana*	[39,51]
isorhoifolin	C96	552-57-8	*S. edule*	[51]
leucoside	C97		*S. edule*	[46]
myricitrin	C98	17912-87-7	*S. edule*	[46]
vicenin 2	C99	23666-13-9	*S. edule*	[46]
all-trans-vaucheriaxanthin	C100		*M. nana*	[69]
neochrome	C101		*M. nana*	[69]
neoxanthin	C102	14660-91-4	*M. nana*	[69]
5,8:5′,8′-diepoxy-5,8,5′,8′-tetrahydro-beta,beta-carotene-3,3′-diol	C103		*M. nana*	[69]
violaxanthin	C104	126-29-4	*M. nana*	[69]
lutein 5,6-epoxide	C105	28368-08-3	*M. nana*	[69]
lutein	C106	127-40-2	*M. nana*	[69]
fumaric acid	C107	110-17-8	*M. nana*	[39]
cis-aconitic acid	C108	585-84-2	*M. nana*	[39]
epigallocatechin	C109	970-74-1	*M. nana*	[39]
3,4-dihydroxybenzoic acid	C110	99-50-3	*M. nana*, *B. oleracea*	[39,40]
protocatechualdehyde	C111	139-85-5	*M. nana*	[39]
epigallocatechin gallate	C112	989-51-5	*M. nana*	[39]
1,5-dicaffeoylquinic acid	C113	30964-13-7	*M. nana*	[39]
4-carboxyphenylglycine	C114	7292-81-1	*M. nana*	[39]
syringic acid	C115	530-57-4	*M. nana*, *B. oleracea*, *F. ananassa*	[39,40,43]
vanillin	C116	121-33-5	*M. nana*	[39]
syringaldehyde	C117	134-96-3	*M. nana*	[39]
daidzin	C118	552-66-9	*M. nana*	[39]
(-)-epicatechingallate	C119	25615-05-8	*M. nana*	[39]
piceid	C120	27208-80-6	*M. nana*	[39]
sinapinic acid	C121	7362-37-0	*M. nana*, *B. oleracea*, *F. ananassa*	[39,40,43]
coumarin	C122	91-64-5	*M. nana*	[39]
querciturone	C123	22688-79-5	*M. nana*	[39]
isoquercetin	C124	482-35-9	*M. nana*	[39]
hesperidin	C125	520-26-3	*M. nana*	[39]
genistin	C126	529-59-9	*M. nana*	[39]
rosmarinic acid	C127	20283-92-5	*M. nana*, *B. oleracea*	[39,40]
apigetrin	C128	578-74-5	*M. nana*	[39]
astragalin	C129	480-10-4	*M. nana*	[39]
kaempferol-3-O-rutinoside	C130	17650-84-9	*M. nana*, *B. oleracea*	[39,40]
fisetin	C131	528-48-3	*M. nana*	[39]
daidzein	C132	486-66-8	*M. nana*	[39]
quercetin 3′-isobutyrate	C133		*M. nana*	[39]
hesperetin	C134	520-33-2	*M. nana*	[39]
genistein	C135	446-72-0	*M. nana*	[39]
kaempferol	C136	520-18-3	*M. nana*	[39]
amentoflavone	C137	1617-53-4	*M. nana*	[39]
cinnamic acid	C138	140-10-3	*B. oleracea*, *F. ananassa*	[40,43]
lactoyl isovanillic acid	C139		*B. oleracea*	[40]
isorhamnetin	C140	480-19-3	*B. oleracea*	[40]
3-hydroxyflavone	C141	577-85-5	*B. oleracea*	[40]
pyrogallol	C142	87-66-1	*B. oleracea*	[40]
pyruvic acid	C143	127-17-3	*B. oleracea*	[40]
lactic acid	C144	50-21-5	*B. oleracea*	[40]
valine	C145	72-18-4	*B. oleracea*	[40]
alanine	C146	56-41-7	*B. oleracea*	[40]
glycolic acid	C147	79-14-1	*B. oleracea*	[40]
linalool, (+/−)-	C148	78-70-6	*F. ananassa*	[70]
beta-Farnesene	C149	18794-84-8	*F. ananassa*	[70]
alpha-Terpineol	C150	98-55-5	*F. ananassa*	[70]
damascenone	C151	23726-93-4	*F. ananassa*	[70]
trans-Nerolidol	C152	35944-21-9	*F. ananassa*	[70]
2,5-dimethyl-4-methoxy-3(2H)-furanone	C153	4077-47-8	*F. ananassa*	[70]
furaneol	C154	3658-77-3	*F. ananassa*	[70]
(E)-5-(3-hexenyl)dihydrofuran-2(3H)-one	C155	97416-87-0	*F. ananassa*	[70]
gamma-Octalactone	C156	104-50-7	*F. ananassa*	[70]
gamma-Decalactone	C157	706-14-9	*F. ananassa*	[70]
gamma-Dodecalactone	C158	148051	*F. ananassa*	[70]
acetophenone	C159	202-708-7	*F. ananassa*	[70]
3,6-Octadienal,3,7-dimethyl-	C160	1754-00-3	*F. ananassa*	[43]
(Z)-3,7-dimethylocta-3,6-dienal	C161	72203-97-5	*F. ananassa*	[43]
citral	C162	C10H16O	*F. ananassa*	[43]
nerylacetone	C163	3879-26-3	*F. ananassa*	[43]
nizatidine	C164	76963-41-2	*F. ananassa*	[43]
beta-caryophyllene oxide	C165	1139-30-6	*F. ananassa*	[43]

Note: “CAS” is the unique number assigned to each chemical substance by the Chemical Abstracts Service. It is used to unambiguously identify chemical compounds in the scientific literature and databases.

## Data Availability

The original contributions presented in the study are included in the article/Appendix A; further inquiries can be directed to the corresponding authors.

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
