# Peer review of "The Discovery of Potential Repellent Compounds for Zeugodacus cucuribitae (Coquillett) from Six Non-Favored Hosts"

_ijms, 2025, doi:10.3390/ijms26146556_

Round 1
Reviewer 1 Report
Comments and Suggestions for Authors
Comments for ijms-3675688:
- The background information provided in the abstract is rather limited. It is recommended that the authors offer a more comprehensive introduction regarding Zeugodacus cucuribitae as an invasive quarantine pest globally, including its impact and the limitations of current control measures. Furthermore, descriptions of the research methods and key findings should be more specific.
- There is a need for further elaboration on the experimental design, criteria for sample selection, and detailed methodologies for data analysis. In particular, additional details and justifications are required to explain how these six plants were identified as less preferred hosts for Z.cucuribitae.
- The manuscript does not delve deeply into the specific mechanisms or effective components of the compounds. It is suggested to conduct additional experiments to validate hypotheses and carry out chemical analyses and identification of potential active ingredients.
- There are several grammatical and spelling errors throughout the text, which detract from the readability and flow of the document.
- Some of the cited references are outdated. Given the rapid advancements in scientific research, it would be beneficial to consult and cite the most recent studies, especially those directly relevant to this work.
- The quality of figures and tables could be improved.
Author Response
|
Comments 1: The background information provided in the abstract is rather limited. It is recommended that the authors offer a more comprehensive introduction regarding Zeugodacus cucuribitae as an invasive quarantine pest globally, including its impact and the limitations of current control measures. Furthermore, descriptions of the research methods and key findings should be more specific. |
|
Response 1: Thank you for pointing this out. We agree with this comment. Therefore,we supplemented the distribution of Zeugodacus cucuribitae in the world, and supplemented the main prevention and control measures of Zeugodacus cucuribitae and the chemical pesticides currently in use. We have marked the revised part in red in the article. The position of the updated data information in the manuscript: page 2, lines 56-67. Page 2, lines 56-67: Zeugodacus cucuribitae (Coquillett) (Z. cucuribitae) is a destructive agricultural pest. it is distributed across a range of climatic regions such as Central and East Asia and Oceania. Due to its vast adaptability, high reproduction potential and invasion capacity, Z. cucuribitae has been the subject of a worldwide pest management program [2].It has widely invaded many crops and brought serious losses to agricultural production. Z. cucuribitae can damage a wide variety of fruits and vegetables, including Luffa acutangular (L. acutangular), Luffa cylindrica (L. cylindrica) and Sechium edule (S. edule). At present, chemical pesticides are one of the main methods to control fruit flies. People mainly use chemical insecticides such as Cypermethrin, Fenvalerate and Abamectin in the field to reduce the damage of fruit and vegetables by Z. cucuribitae. However, these methods have some limitations, such as chemical pesticides polluting the environment and limited physical trapping effect.
|
|
Comments 2:There is a need for further elaboration on the experimental design, criteria for sample selection, and detailed methodologies for data analysis. In particular, additional details and justifications are required to explain how these six plants were identified as less preferred hosts for Zeugodacus cucuribitae. |
|
Response 2: Thank you for pointing this out. We agree with this comment. Therefore, We analyzed the preference of these Zeugodacus cucuribitae for these hosts through some reported literature data. The results showed that, compared with other hosts, Zeugodacus cucuribitae was more sensitive to Luffa acutangular, Luffa cylindrica, Sechium edule, and Brassica oleracea var. botrytis. The visit rate and oviposition rate, as well as the relative host performance and fecundity of Musa nana and Fragaria × ananassa, were lower. The position of the updated data information in the manuscript: page 2-3 lines 88-94. Page 2-3 lines 88-94: The main host of Z. cucuribitae is Cucurbitaceae, however, L. acutangular, L. cylindrica and S. edule are relatively no much favourite host of Z. cucuribitae among Cucurbitaceae, Because compared with other cucurbitaceae plants, the visit rate and oviposition rate of these species by Z. cucuribitae are lower [16-18]. In addition, according to the report, some non-cucurbitaceae plants, such as Brassica oleracea var. botrytis (B. oleracea), Musa nana (M. nana) and Fragaria × ananassa (F. ananassa), are relatively less attractive to Z. cucuribitae, The Relativehost performance and Fecundity of Z. cucuribitae to these species is significantly lower than that of other species. |
|
Comments 3: The manuscript does not delve deeply into the specific mechanisms or effective components of the compounds. It is suggested to conduct additional experiments to validate hypotheses and carry out chemical analyses and identification of potential active ingredients. |
|
Response 3: Thank you for pointing this out. We agree with this comment. Therefore,we will make a prediction through the Mechanism of the effective components verified by the Gastric toxicity test and the Two-way selection experiment. By constructing composite pathways, the key targets and pathways in upstream and downstream can be predicted, so as to predict the avoidance behavior or death mechanism of effective components to Zeugodacus cucuribitae. Further, we added corresponding contents in the abstract, materials and methods, results and analysis and discussion. We have marked the revised part in red in the article. The position of the updated data information in the manuscript: page 1-2 lines 45-48; page 13-14 lines 307-332; page 15 lines387-407; page 18 lines 525-536. Page 1-2 lines 45-48: By Kegg enrichment and molecular mechanism derivation of the genes successfully docked with these components, we found that these components probably played their roles through neuroactive ligand-receptor interaction(ko04080) and Calcium signaling pathway(ko04020). Page 13-14 lines 307-332: 3.6. Predicted Mechanism of Core Components Through indoor experiments, we verified the effective effects of oleanolic acid, rotenone, and beta-caryophyllene oxide on Z. cucurbitae. The control effects of these components on Z. cucurbitae were achieved through the joint action of multiple genes and pathways. According to the complex network, the targets of the successful actions of these components can be identified. Through the CGGP network constructed, we identified the gene LOC105217288. After KEGG enrichment of LOC105217288, it was found that these components were mainly enriched in two pathways: neuroactive ligand-receptor interaction (ko04080) and calcium signaling pathway (ko04020). By marking the activated genes in these two pathways, we deduced a molecular mechanism of the influence of components apigenin, testosterone propionate, and biochanin A on the behavior of Z. cucurbitae. These components further activate proteins located on the cell membrane by activating the gene LOC105217288, including glutamate receptor ionotropic NMDA1 (GRIN), glutamate receptor 1 (GRI), and nicotinic acetylcholine receptor alpha-7 (nAChRα7). These targets further affect downstream pathways by activating calcium ions and calmodulin (CALM), such as the MAPK signaling pathway (ko04010), apoptosis (ko04210), and long-term depression (ko04730). Eventually, Z. cucurbitae exhibits avoidance behavior and even death. According to the report, After studying the influence of Chironomus dilutus on imidacloprid, it is found that neonicotinoids can activate neuroactive ligand-receptor interaction (ko 04080), which leads to the dysfunction of insect nervous system, and then lead to oxidative stress, mitochondrial dysfunction, inflammatory reaction and apoptosis. By combining RNA-Seq with isobaric ectopic tags for relative and absolute quantitative (iTRAQ) analysis, it was found that calcium signaling pathway (ko 04020) played an important role in the adaptation and toxic reaction of Bursaphelenchus mucronatus to the host.
Fig. 6. Mechanism deduction of the effect of effective components on odor recognition of Z.cucurbitae. Red nodes represent activated genes. Page 15 lines387-407: After verifying the effects of oleanolic acid, rotenone, and beta-caryophyllene oxide on Z. cucurbitae, the gene LOC105217288 was identified through KEGG enrichment of the gene set successfully docked with these components. Function prediction was carried out by constructing a composite pathway, and it was found that LOC105217288 is mainly involved in neuroactive ligand-receptor interaction (ko04080) and calcium signaling pathway (ko04020). In these pathways, the proteins glutamate receptor ionotropic NMDA1 (GRIN), glutamate receptor 1 (GRI), and nicotinic acetylcholine receptor alpha-7 (nAChRα7) are activated, thereby affecting downstream proteins and pathways. For example, calmodulin (CALM) is activated, which influences pathways such as the MAPK signaling pathway (ko04010), apoptosis (ko04210), and long-term depression (ko04730). Neuroactive ligand-receptor interaction (ko04080) and calcium signaling pathway (ko04020) have been widely reported in insect research and are closely related to insect avoidance and death. Additionally, the activation of these pathways can indirectly affect downstream pathways, such as the MAPK signaling pathway (ko04010), apoptosis (ko04210), and long-term depression (ko04730). Transcriptome analysis showed that the expression of the MAPK signaling pathway (ko04010) changed after Bombyx mori was parasitized by Exorista japonica. Studies show that MAPK signaling pathway plays an important role in the response of insects to parasitic stress. Transcriptome analysis and PCR-RFLP analysis showed that the differential genes of Aphis gossypii and Aedes aegypti after being treated with pyrethroid insecticides were closely related to the genes in the apoptosis pathway (ko04210). Page 18 lines 525-536: 4.6.. Predicted Mechanism of Core Components The verified effective components and the genes that were successfully docked in the CPPG network were collected. Then, after sorting out this gene information and the corresponding enrichment information, we accessed the KEGG database, accurately mapped it to the corresponding KEGG pathway map, and recorded the position of each gene in the pathway in detail. On this basis, combined with the genes and products of upstream and downstream pathways, we constructed the hypothesis of the visual action mechanism of composite pathways. Furthermore, we deeply analyzed the overall biological function of the KEGG pathways labeled with genes, clarified their main roles in physiological processes, and, in combination with the positions of labeled genes, discussed the influence of metabolic reactions or signal transduction events on the operation of the entire pathway. |
|
Comments 4: There are several grammatical and spelling errors throughout the text, which detract from the readability and flow of the document. |
|
Response 4: Thank you for pointing this out. We agree with this comment. Therefore,We checked the grammar of the manuscript, corrected the grammatical errors in the manuscript and made the sentence more suitable for reading. |
|
Comments 5: Some of the cited references are outdated. Given the rapid advancements in scientific research, it would be beneficial to consult and cite the most recent studies, especially those directly relevant to this work |
|
Response 5: Thank you for pointing this out. We agree with this comment. Therefore,We supplement some recent research progress to testify our conclusions in the study, so as to ensure that the references are more timely and relevant. These updates not only enrich the literature review, but also further support our research viewpoint. We have marked the revised part in red in the article. The position of the updated data information in the manuscript: page 20, lines 579-581; page 21, lines 623-630; page 22, lines 653-656; page 26, lines 835-837 Page 20, lines 579-581: 3. Singh, Ram. "M.K. Dhillon, Ram Singh, J.S. Naresh and H.C. Sharma. 2005. The Melon Fruit Fly, Bactrocera Cucurbitae: A Review of Its Biology and Management. Journal of Insect Science (USA) 5: 40.(7.45)." Journal of Insect Science 5 (2022): 40. page 21, lines 623-630: 18. Zeng, Bei, Yuyang Lian, Jingjing Jia, Yang Liu, Aqiang Wang, Heming Yang, Jinlei Li, Shuyan Yang, Sihua Peng, and Shihao Zhou. "Multigenerational Effects of Short-Term High Temperature on the Development and Reproduction of the Zeugodacus Cucurbitae (Coquillett, 1899)." Agriculture 12, no. 7 (2022): 954. 19. Bragard, C., K. Dehnen-Schmutz, F. Di Serio, P. Gonthier, M. A. Jacques, J. A. Jaques Miret, A. F. Justesen, A. MacLeod, C. S. Magnusson, P. Milonas, J. A. Navas-Cortes, S. Parnell, R. Potting, P. L. Reignault, H. H. Thulke, A. Vicent Civera, J. Yuen, L. Zappalà, N. Papadopoulos, S. Papanastasiou, E. Czwienczek, V. Kertész, and A. MacLeod. "Scientific Opinion on the Import of Musa Fruits as a Pathway for the Entry of Non-Eu Tephritidae into the Eu Territory." Efsa j 19, no. 3 (2021): e06426. Page 22, lines 653-656: 28. Wang, Jing jing, Chao Ma, Zhen ya Tian, Yong ping Zhou, Jin fang Yang, Xuyuan Gao, Hong song Chen, Wei hua Ma, and Zhong shi Zhou. "Electroantennographic and Behavioral Responses of the Melon Fly, Zeugodacus Cucurbitae (Coquillett), to Volatile Compounds of Ridge Gourd, Luffa Acutangular L." Journal of Chemical Ecology 50, no. 12 (2024): 1036-45. Page 26, lines 852-854: 97. Dai, Minli, Jin Yang, Xinyi Liu, Haoyi Gu, Fanchi Li, Bing Li, and Jing Wei. "Parasitism by the Tachinid Parasitoid Exorista Japonica Leads to Suppression of Basal Metabolism and Activation of Immune Response in the Host Bombyx Mori." Insects 13, no. 9 (2022): 792.
|
|
Comments 6: The quality of figures and tables could be improved |
|
Response 6: Thank you for pointing this out. We agree with this comment. Therefore,We have updated the expression of some figures. Figure 1 now shows only the phylogenetic analysis of the IR gene, The phylogenetic analysis of all genes was added to the supplementary materials of the manuscript as additional content. Figures 3A, 3B, and 5 use brighter colors to make the data more clearly visible. To address the label overlapping phenomenon of some nodes in Figure 4, we have reduced the number of gene and component nodes to 30, making the labels of each node more clearly visible. We have replaced the revised pictures in the manuscript. The position of the updated figures in the manuscript:
Figure 1. Construction of Phylogenetic analysis of IR genes based on gene sequences of Z. cucuribitae and several Diptera insects. Notes: Acer_Apis Cerana,Bcuc:Zeugodacus Cucurbitae,Bdor:Bactrocera Dorsalis,Bole:Bactrocera Oleae,Dmel:Drosophila Melanogaster,Mdom:Musca Domestica,Dana:Drosophila Ananassae,Dsim:Drosophila Simulans.
Figure 3. Visualization of docking of olfactory sensory genes with compounds of hosts. (A) The numbers of compounds with different affinities between -5 kcal/mol and18 kcal/mol were counted. (B) The numbers of genes with different affinities between -5 kcal/moland -18 kcal/mol were counted. (C) Prediction results of molecular docking between Maslinic acid 3-O-b-D-glucoside and gene LOC105217972 of Luffa cylindrica.; (D) Prediction of the combination of the metabolic component Echinocystic acid and the gene LOC105218953 of Luffa cylindrica; (E) Prediction of the combination of Isorhoifolin and LOC105215418; (F) Prediction of the combination of Kaempferol, a metabolic component of banana, and LOC105209256; (G) Prediction of the combination of Rutin, a metabolic component of cauliflower, and LOC105209182; (H) Prediction of the combination of Damascenone, a metabolic component of strawberry, and LOC105217972.
Figure 4. Complex network node diagram of components of L. acutangular, L. cylindrica, S. edule, B. oleracea , M. nana and F. ananassa vs. Z. cucuribitae olfactory sensory genes vs Kegg pathway vs. GO- term. The core components are represented in blue, the Z. cucuribitae olfactory sensory genes are represented in yellow, the Kegg pathway is represented in green, and the GO- term is represented in red. The metabolites selected for two-way selection test and gastric toxicity test are shown in dark blue.
Figure 5. Avoidance rate and mortality rate of Zeugodacus cucuribitae under different concentrations of five metabolic components. (A) Avoidance rate of Rotenone. (B) Avoidance rate of Diosmin. (C) Avoidance rate of beta-Caryophyllene Oxide.(D) Avoidance rate of Oleanolic Acid. (E) Avoidance rate of Echinocystic acid.(F) mortality rate of Rotenone. (G) mortality rate of Diosmin. (H) mortality rate of beta-Caryophyllene Oxide.(I) mortality rate of Oleanolic Acid.(J) mortality of Echinocystic acid. ns means no significance between the compared two groups.; * represents Significant different in the level of p < 0.05; ** represents Significant different in the level of p < 0.01; *** represents Significant different in the level of p < 0.001 compared to the model group.
|

Reviewer 2 Report
Comments and Suggestions for Authors
The manuscript, submitted by Y. Chen, S. Jin, Z. Zhou, and co-authors, reports on the discovery of some potential repellent compounds on Zeugodacus cucuribitae.
In general, it is a manuscript that is a pleasure to read: the Introduction provides all necessary background information, the Materials and Methods paragraph seems to be generally well-organized, and the Discussions paragraph is simple and convincing.
In my honest opinion, the paper should be considered for publication in the International Journal of Molecular Sciences, after some changes and revisions, which can be considered as minor:
- Abstract: the authors claim that ‘Z. c. can harm more than 180 kinds of fruits and vegetables’. Why exactly 180? I agree that the species is extremely invasive and can harm many domestic species, but why exactly 180??
- The adaptation of various plants to evolve means, which are helping them to avoid harm from insects, is indeed amazing. However, the authors should explain the meaning of the ‘plant insect-resistant strategy’ – phrase.
- Paragraph 2.1.: the authors should use a blank space when writing masses etc. g.: ‘50g’ should read 50 g. Please, check throughout the paper.
- Discussions: once again, we have the famous ‘180 kids of crops’. I suggest replacing this phrase with a simpler ‘more than 100’, many’, etc.
- Minor typesetting remark: in Discussions, Paragraph 2, ‘In this study, The metabolic …’ should read: ‘In this study, the metabolic …’
- Another minor suggestion: in the Discussions, avoid starting two consecutive paragraphs with the same phrase: ’In this study, (…)’.
Comments on the Quality of English Language
The English could be improved to more clearly express the research.
Author Response
|
Comments 1: Abstract: the authors claim that ‘Z. c. can harm more than 180 kinds of fruits and vegetables’. Why exactly 180? I agree that the species is extremely invasive and can harm many domestic species, but why exactly 180?? |
|
Response 1: Thank you for pointing this out. We agree with this comment. Therefore,In order to be rigorous and scientific, we changed the original expression “Z.cucuribitae can harm more than 180 fruits and vegetables, including Luffa acutangular (L.acutangular), Luffa cylindrica (L.cylindrica), Sechium edule (S.edule)” to: “Z. cucuribitae can damage a wide variety of fruits and vegetables, including Luffa acutangular (L. acutangular), Luffa cylindrica (L. cylindrica) and Sechium edule (S. edule)”. This makes the description of the host of Z.cucuribitae more accurate, and further, we also modify the same expression in the discussion part. We have marked the revised part in red in the article. The position of the updated data information in the manuscript: page 2, lines 60-61; page 14, lines 335-337. Page 2, lines 60-61 Z. cucuribitae can damage a wide variety of fruits and vegetables, including Luffa acutangular (L. acutangular), Luffa cylindrica (L. cylindrica) and Sechium edule (S. edule). Page 14, lines 335-337: Z. cucurbitae is a destructive agricultural pest that can damage a wide variety of fruits and vegetables, including L. acutangular, L. cylindrica and S. edule, resulting in the destruction of fruits and the decline of yield and quality. |
|
Comments 2: The adaptation of various plants to evolve means, which are helping them to avoid harm from insects, is indeed amazing. However, the authors should explain the meaning of the ‘plant insect-resistant strategy’ – phrase. |
|
Response 2: Thank you for pointing this out. We agree with this comment. Therefore,After a rigorous examination and access to relevant reports, we confirmed that AAA is an incorrect term, and we have revised it in the manuscript and described it more rigorously. We have marked the revised part in red in the article. The position of the updated data information in the manuscript: page 2, lines 72-73 Page 2, lines 72-73: Reducing the damage of insects through secondary metabolites is the main means for plants to fight pests. |
|
Comments 3: Paragraph 2.1.: the authors should use a blank space when writing masses etc. g.: ‘50g’ should read 50 g. Please, check throughout the paper. |
|
Response 3: Thank you for pointing this out. We agree with this comment. Therefore,We corrected the errors caused by not adding spaces between numbers and units, and checked and corrected similar errors by checking the full text. We have marked the revised part in red in the article. The position of the updated data information in the manuscript: page 14, lines 411-413. Page 14, lines 411-413: The larvae of the Z. cucuribitae colony were provided with an artificial larval diet mixture of 50 g of yeast extract, 250 g of wheat bran powder, 50 g of sugar, 1 g of sodium benzoate, 50 g of paper and 400 mL of water. |
|
Comments 4: Discussions: once again, we have the famous ‘180 kids of crops’. I suggest replacing this phrase with a simpler ‘more than 100’, many’, etc. |
|
Response 4: Thank you for pointing this out. We agree with this comment. Therefore,In order to be rigorous and scientific, we changed the original expression “Z.cucuribitae is a destructive agricultural pest, which widely harms 180 kinds of crops such as bitter gourd and zucchini, resulting in the destruction of fruits and the decline of yield and quality.” to: “Z. cucurbitae is a destructive agricultural pest that can damage a wide variety of fruits and vegetables, including L. acutangular, L. cylindrica and S. edule, resulting in the destruction of fruits and the decline of yield and quality.”. We have marked the revised part in red in the article. The position of the updated data information in the manuscript: page 14, lines 335-337. Page 14, lines 335-337: Z. cucurbitae is a destructive agricultural pest that can damage a wide variety of fruits and vegetables, including L. acutangular, L. cylindrica and S. edule, resulting in the destruction of fruits and the decline of yield and quality. |
|
Comments 5: Minor typesetting remark: in Discussions, Paragraph 2, ‘In this study, The metabolic …’ should read: ‘In this study, the metabolic …’ |
|
Response 5: Thank you for pointing this out. We agree with this comment. Therefore,We revised the discussion on the misuse of uppercase and lowercase, and corrected similar mistakes by checking the full text. We have marked the revised part in red in the article. The position of the updated data information in the manuscript: page 17, lines 473-474. Page 17, lines 473-474: The metabolic components of L. acutangular, L. cylindrica, S. edule, B. oleracea, M. nana and F. ananassa were collected from the literature. |
|
Comments 6: Another minor suggestion: in the Discussions, avoid starting two consecutive paragraphs with the same phrase: ’In this study, (…)’. |
|
Response 6: Thank you for pointing this out. We agree with this comment. Therefore,We modified the expression of the whole discussion, and reduced repetitive sentence patterns through diversified expressions, increased the fluency of reading, and made the manuscript still rigorous and scientific. We have marked the revised part in red in the article. The position of the updated data information in the manuscript: page 14-15, lines 335-407. Page 14-15, lines 335-407: Z. cucurbitae is a destructive agricultural pest that can damage a wide variety of fruits and vegetables, including L. acutangular, L. cylindrica and S. edule, resulting in the destruction of fruits and the decline of yield and quality. Currently, the main control methods for Z. cucurbitae are chemical pesticides, which, however, have obvious limitations. For example, deltamethrin and abamectin have been reported to have limited effectiveness [3, 4, 7]. Previous studies have shown that plant metabolites have good control effects and economic value in pest control. For instance, azadirachtin isolated from Azadirachta indica has been proven effective [91, 92]. Host plants also contain insect-resistant metabolites to defend against insect damage [93-95]. Long-term observations have indicated that although Z. cucurbitae can harm L. acutangular, L. cylindrica, S. edule, B. oleracea, M. nana, and F. ananassa, the damage is not severe, a phenomenon that has been previously reported [96]. It is believed that these plants produce defensive metabolites that reduce pest damage. It is speculated that both cucurbitaceous hosts, such as L. acutangular, L. cylindrica, and S. edule, and non-cucurbitaceous hosts, such as B. oleracea, M. nana, and F. ananassa, contain metabolites that adversely affect Z. cucurbitae, thereby reducing the damage caused by this pest. These potential metabolites are worthy of further exploration and study for the prevention and control of Z. cucurbitae. The metabolic components of L. acutangular, L. cylindrica, S. edule, B. oleracea, M. nana and F. ananassa were collected from the literature. Meanwhile, the olfactory sensory genes of Z. cucurbitae were gathered based on its genome. Molecular docking was then conducted between these metabolic components and the olfactory sensory genes. Moreover, the genes were enriched via KEGG pathway and GO-term analyses, and a CPPG network was constructed. Through these steps, important components that may have potential effects on Z. cucurbitae were identified. In the two-way selection experiment, oleanolic acid (1 mg/mL, 0.1 mg/mL, 0.01 mg/mL), rotenone (1 mg/mL, 0.1 mg/mL, 0.01 mg/mL), and beta-caryophyllene oxide (1 mg/mL, 0.1 mg/mL) were successfully screened. In the gastric toxicity experiment, we found that echinocystic acid (1 mg/mL, 0.1 mg/mL, 0.01 mg/mL), rotenone (1 mg/mL, 0.1 mg/mL), and beta-caryophyllene oxide (1 mg/mL, 0.1 mg/mL) had significant gastric toxicity to Z. cucurbitae. In the two-way selection experiment, three metabolites—echinocystic acid (1 mg/mL, 0.1 mg/mL, 0.01 mg/mL), rotenone (1 mg/mL, 0.1 mg/mL), and beta-caryophyllene oxide (1 mg/mL, 0.1 mg/mL)—which have significant repellent effects and insecticidal activity against Z. cucurbitae, were successfully screened. Among these components, rotenone is a widely used botanical insecticide, which has various insecticidal activities including neurotoxicity against Spodoptera litura and Bombus terrestis [86, 87]. beta-Caryophyllene Oxide has also been widely reported to have significant toxicity to Sitophilus Granarius L. and Callosobruchus chinensis[81-83]. Oleanolic Acid was found to have significant antifeedant activity against Aedes aegypti L. and spodoptera litura F. [84, 85]. However, there is still a alack in the study of echinocystic acid in controlling pest, indicating that this component was worth to be further study. In this study, we found that rotenone, oleanolic acid, beta-caryophyllene oxide, and echinocystic acid had significant repellent or stomach toxic effects on Z. cucurbitae. These findings can be used as a reference for the prevention and control of Z. cucurbitae. Not only were the metabolites rotenone, oleanolic acid, beta-caryophyllene oxide, and echinocystic acid with significant repellent effects screened from the hosts of Z. cucurbitae, but their repellent and insecticidal activities against Z. cucurbitae were also verified for the first time. This lays a solid foundation for further studies on the repellent mechanisms of Z. cucurbitae and the development of new and efficient control strategies. Future research will further explore the mechanisms, optimization, and applications of these metabolites to provide a scientific reference for the effective prevention and control of Z. cucurbitae. After verifying the effects of oleanolic acid, rotenone, and beta-caryophyllene oxide on Z. cucurbitae, the gene LOC105217288 was identified through KEGG enrichment of the gene set successfully docked with these components. Function prediction was carried out by constructing a composite pathway, and it was found that LOC105217288 is mainly involved in neuroactive ligand-receptor interaction (ko04080) and calcium signaling pathway (ko04020). In these pathways, the proteins glutamate receptor ionotropic NMDA1 (GRIN), glutamate receptor 1 (GRI), and nicotinic acetylcholine receptor alpha-7 (nAChRα7) are activated, thereby affecting downstream proteins and pathways. For example, calmodulin (CALM) is activated, which influences pathways such as the MAPK signaling pathway (ko04010), apoptosis (ko04210), and long-term depression (ko04730). Neuroactive ligand-receptor interaction (ko04080) and calcium signaling pathway (ko04020) have been widely reported in insect research and are closely related to insect avoidance and death. Additionally, the activation of these pathways can indirectly affect downstream pathways, such as the MAPK signaling pathway (ko04010), apoptosis (ko04210), and long-term depression (ko04730). Transcriptome analysis showed that the expression of the MAPK signaling pathway (ko04010) changed after Bombyx mori was parasitized by Exorista japonica. Studies show that MAPK signaling pathway plays an important role in the response of insects to parasitic stress[97]. Transcriptome analysis and PCR-RFLP analysis showed that the differential genes of Aphis gossypii and Aedes aegypti after being treated with pyrethroid insecticides were closely related to the genes in the apoptosis pathway (ko04210)[98, 99]. In the conclusion, not only through the combination of network pharmacology and molecular docking technology, but also based on the comprehensive analysis of the metabolic components of L. acutangular, L. cylindrica, S. edule, B. oleracea, M. nana, and F. ananassa and the olfactory sensory genes of Z. cucurbitae, a CPPG network was established. The metabolites rotenone, oleanolic acid, beta-caryophyllene oxide, and echinocystic acid, which have significant repellent effects, were successfully screened. The repellent and insecticidal activities of these components against Z. cucurbitae were also verified for the first time. This lays a solid foundation for further studies on the repellent mechanisms of Z. cucurbitae and the development of new and efficient control strategies. Future research will further explore the mechanisms, optimization, and applications of these metabolites to provide a scientific reference for the effective prevention and control of Z. cucurbitae.
|
